# Femtosecond Laser-Induced Nano-Joining of Volatile Tellurium Nanotube Memristor

**DOI:** 10.3390/nano13050789

**Published:** 2023-02-21

**Authors:** Yongchao Yu, Pooran Joshi, Denzel Bridges, David Fieser, Anming Hu

**Affiliations:** 1Department of Mechanical, Aerospace and Biomedical Engineering, University of Tennessee Knoxville, 1512 Middle Drive, Knoxville, TN 37996, USA; 2School of Mechanical and Aerospace Engineering, Nanyang Technological University, 50 Nanyang Ave., Singapore 639798, Singapore; 3Oak Ridge National Lab, 1 Bethel Valley Rd., Oak Ridge, TN 37831, USA

**Keywords:** memristor, tellurium nanotube, laser joining, nano-joining

## Abstract

Nanowire/nanotube memristor devices provide great potential for random-access high-density resistance storage. However, fabricating high-quality and stable memristors is still challenging. This paper reports multileveled resistance states of tellurium (Te) nanotube based on the clean-room free femtosecond laser nano-joining method. The temperature for the entire fabrication process was maintained below 190 °C. A femtosecond laser joining technique was used to form nanowire memristor units with enhanced properties. Femtosecond (fs) laser-irradiated silver-tellurium nanotube-silver structures resulted in plasmonic-enhanced optical joining with minimal local thermal effects. This produced a junction between the Te nanotube and the silver film substrate with enhanced electrical contacts. Noticeable changes in memristor behavior were observed after fs laser irradiation. Capacitor-coupled multilevel memristor behavior was observed. Compared to previous metal oxide nanowire-based memristors, the reported Te nanotube memristor system displayed a nearly two-order stronger current response. The research displays that the multileveled resistance state is rewritable with a negative bias.

## 1. Introduction

With the development of artificial neural networks, neuromorphic computation has become a promising and highly desired technology in big data analysis, artifice intelligence, and next-generation computation [1]. Traditional Si-based and complementary metal-oxide semiconductor (CMOS) computation technology requires separated arithmetic/logic and memory units. Neuromorphic computation aims to provide simultaneous processing memory in one unit without the burden of data transmission, which targets significantly higher efficiency and superior computational speed than conventional Von Neumann architectures. In recent years, hardware-implemented neural network computation systems have attracted much interest due to their possible machine learning applications with low power consumption [2,3]. Conventionally, transistor devices are used as a calculation unit to emulate voltage-dependent communication across biological synapses. However, the three-terminal structure of the semiconductive transistors increases the difficulty of large-scale integration and results in higher power consumption than two-terminal devices [4,5].

Since the Hewlett Packard lab produced the first memristor in 2008, it has become a breakthrough component for neuromorphic circuits [6,7]. As a two-terminal memory device, it has been considered an ideal device for simulating a synapse of neural network systems [8,9,10]. Memristors can be categorized into volatile and nonvolatile types based on the retention time of the low-resistance state. It can maintain the high-resistance state (HRS) and low-resistance state (LRS) after the bias voltage is removed. On the other hand, volatile memristors would spontaneously return to HRS in milliseconds to nanoseconds after removal of an external excitation. The switching mechanisms of memristors have been reported, including a Schottky barrier, Pool–Frenkel emission, space-charge-limited current, trap charging and discharging, ferroelectric polarization, electron spin, growth and fracture of conductive filaments, etc. For a detailed discussion of the switching mechanism, Sun et al., Guo et al., and Wang et al. have recently provided comprehensive reviews [11,12,13,14].

Volatile memristors have garnered increasing attention in recent years due to their potential applications in mimicking synaptic functions and serving as ideal sources of randomness for entropy. The volatile memristors can be divided into threshold switching (digital switching) and analog switching according to the switching type. An abrupt switching behavior and a large ON/OFF ratio are characteristics of the threshold switching memristors. When the applied voltage surpasses the threshold voltage, the current response of the memristor sharply increases. Similarly, when the applied voltage is lower than the hold voltage, the current response of the memristor decreases abruptly. In addition, the threshold switching occurs in the nano-to-microsecond period with a large ON/OFF ratio [15]. Therefore, the threshold switching memristors are considered a good candidate for the access device and logic circuits.

On the other hand, the volatile memristors with analog switching feature continuous and incremental resistance changes. The ON/OFF ratio of the analog switching memristors is relatively lower than that of threshold-type memristors. In the past few decades, a great number of volatile memristors have been developed, as shown in Table 1. For more applications and recent development of volatile memristors, Wang et al. provided a detailed review [15].

Till now, many of these applications have been focused on thin film metal-oxide semiconductive materials [25,26,27,28] because the structure and fabrication process of thin film memristors are very similar to the CMOS devices. Metal-Oxide-Metal (MOM) is the most typical structure; An active layer is sandwiched between two contact electrodes to form a sandwich structure. Such similarity significantly accelerates the development of the memristor.

However, the size of the contact electrodes for MOM structure usually is relatively large (in a µm to mm range). The resistance stage switching process, such as growth and fracture of conductive filaments, phase shift of active material, and migration of ions/oxygen vacancy/protons, can occur anywhere in the electrode area with certain randomness. This randomness is uncontrollable and unpredictable, which leads to increase of instability in device-to-device and cycle-to-cycle [29]. Such characteristics have complicated the understanding of the physical and dynamic molding of memristors. In recent years, scholars have found that reducing the size of electrodes for memristors and scale down the film thickness can effectively reduce such randomness and increase stability [30,31].

Single nanowire/nanotube devices are currently considered ideal devices to deeply understand the physical and electrochemical mechanism of memristors [32]. In a single nanowire/nanotube memristor devise, the electrochemical reactions of the switching process can be controlled within the range of the nanowire diameter (usually less than 200 nm). The highly concentrated and localized electrochemical reaction can effectively control and reduce the randomness [29]. Simultaneously, the single nanowire/nanotube device has a simple 2D structure, which simplifies fabrication, observation, and characterization. The single nanowire/nanotube memristor is of great help in further studying the switching mechanism of different materials and establishing the dynamics model of the memristors.

However, properly connecting nanowires/nanotubes to the electrodes is a critical problem in nanowire/nanotube-based functional devices. The interface between electrodes and nanowires can determine the overall performance of the functional devices. Joining nanowires/nanotubes to the electrodes by using focused ion beam deposition is an accurate and reliable method but very costly [33]. Other techniques, such as chemical processing and photolithography, are either complicated or have weak operation accuracy [34,35,36]. In a mountain of methods, femtosecond laser nano-joining is an alternative joining method and shows a promising result for joining nanostructures [37]. The minimized thermal-affected zone induced by femtosecond laser enables highly localized joining. Several researchers have used fs lasers for interface processing of nanowires/nanotubes on electrodes and further modified the performance of the function devices, such as transistors and memristors [38,39,40]. Xiao et al. presented a comprehensive review of laser-induced joining of nanoscale materials [41]. Moreover, laser post-treatment is very promising for modifying the performance of the memristors. Koryazhkina et al. reported that using laser and thermal treatment can significantly enhance the hysteresis loop in I–V curves of SiNx-based memristors and reduce switching voltage [42]. The effects were attributed to the change of positive-charge density and the annealing of surface states. Ghasemi et al. reposted that the laser-annealing results in the forming of functional thin memristive layers from metallic materials by oxidation of the Ni_80_Fe_20_ (Permalloy, Py) layer and the forming of a uniform Py/Py-oxide heterostructure [43].

Tellurium has gained considerable interest as the active material in transistors in these years [44,45], but its memristive behavior has yet to be widely reported. In this study, we report a single tellurium nanotube-based volatile memristor for the first time. This work shows the practicability and effectiveness of using tellurium nanotube/nanowire as the memristor device. The device shows a volatile memristor behavior with multiple resistance states. Additionally, the experiment revealed that the fs irradiation process could adjust memristor characteristics and increase the performance and stability of the memristor.

## 2. Experiments, Measurements Setup, and Device Fabrication

### 2.1. Nanotube Synthesis

Tellurium nanotubes (TeNT) were synthesized by a hydrothermal process. First, 0.1 mmol TeO_2_ (Sigma-Aldrich Pte. Ltd., St. Louis, MO, USA, purity ≥ 99%) and 0.23 g cetyltrimethylammonium bromide (CTAB, Fisher Chemical™, Waltham, MA, USA, purity 98%) were added into a 25 mL Teflon-lined stainless-steel autoclave. Then, the autoclave was filled with Ethylene glycol (Fisher Chemical™) to 80% of the total volume of the autoclave. It was heated to 180 °C for 24 h in a furnace (Quincy Lab, Inc. Burr Ridge, IL, USA, Model 10AF) and then naturally cooled to room temperature. Silver-gray solids were collected after centrifuging (ThermoFisher Scientific, Waltham, MA, USA) the reacted mixture at 5000 rpm for 30 min. CTAB residuals were removed by washing the particles several times, alternating between distilled water and ethanol. Appendix A shows an SEM image of synthesized TeNTs. The average length of the nanotubes is about 20 µm, and the average outer diameter is approximately 500–600 nm (as shown in Appendix A). The wall thickness of the nanotubes, measured based on SEM images and ImageJ software (Version 1.53t), is about 30–50 nm. Appendix A presents an X-ray diffraction (XRD) analysis (Empyrean). The XRD analysis reveals minimal impurities, which confirms that high-purity TeNTs have been synthesized.

### 2.2. Silver Electrode Fabrication, Nanotube Manipulation, and fs Laser Joining

Silver electrodes were printed on photo paper by an Optomec AJ200 aerosol jet printer (Appendix A). A silver nanoparticle ink (ClariantPte Ltd., Singapore, TPS 50G2) was used in this experiment. The liquid-phased silver ink was atomized and created a dense aerosol composed of droplets. The diameters of the droplets are approximately 1–5 µm. The atomized droplets were carried to the deposition head by N_2_ gas flow. Therewith, the droplets were further concentrated by the specially designed structure at the depositing head. Depending on the size of the nozzles, the width of the printed line is in the range of 10 µm to 1 mm. In this experiment, a 300 µm nozzle was used during the printing process. The silver electrodes were printed by 80 µm printing paths with a 35% overlap. The image of the printed silver electrodes is presented in Appendix A. The thickness of the printed silver electrodes is around 2–3 µm, as shown in Appendix A. The printed silver electrodes were cured in a furnace at 125 °C for 60 min at air environment and under atmospheric pressure. After curing, the silver electrodes show a good conductivity. The surface morphology of cured silver electrodes is shown in Appendix A. The sheet resistance was characterized by a standard four-point measurement method. The sheet resistance of the printed silver electrodes is 155 mΩ/□.

A fs laser processing system was developed for laser-joining processing and creating an interdigit structure. A fs laser (Calmer Laser, Inc., Palo Alto, CA, USA), at a wavelength of 1030 nm, a pulse width of 350 femtoseconds, and a repetition rate of 120 kHz, with a variable power range of 0–1.5 W, was used to cut a gap on the printed electrodes to form the interdigitated structure. The laser beam was focused using a long working distance 100× microscope lens (Numerical aperture NA: 0.8) on the photo paper surface. A schematic of the laser processing system is shown in Appendix A. The gap size of the interdigit structure was kept at about 5–8 µm (as shown in Appendix A).

In order to fabricate a nanotube-based device, the nanotubes need to be vertically aligned with the electrodes. Figure 1a shows Te nanotubes randomly distributed in the vicinity of the silver film gap generated by the focused laser beam. To properly manipulate the nanotube, the dielectrophoretic process (DEP) was applied to manipulate nanotubes. Deionized water-diluted TeNT solution was dropped on the electrode surface, and 5 V peak-to-peak AC current was applied to the electrode. The AC frequency was set to 5 MHz. When nanotubes are subjected to a nonuniform electric field, the nanowires/nanotubes will be polarized and aligned with a minimum energy configuration [46]. The nanowires or nanotubes can be transversely aligned with the gap. The alignment result is affected by various other factors, such as the suspending medium, dielectric constant, and the geometry of nanowires/nanotubes. Figure 1b shows a scanning electron microscope (SEM) image of the nanotubes manipulated by the DEP process. As shown in the figure, TeNTs bridge the gap and connect two sides of the silver electrodes after DEP manipulation. After DEP alignment, the samples were placed under a microscope for observation. A single nanotube that was vertically aligned with the gap was chosen, and all other nanotubes were cut by irradiating a laser at high power (100 mW).

Figure 1c shows an SEM image of the TeNT before fs laser irradiation. The TeNT attaches to the silver electrode without any physical bonding. To improve the electrical connection between TeNT and Ag electrodes, the fs laser was focused on the interface between the nanotubes and the silver electrodes. The diameter of the focused laser beam was approximately one wavelength (~1 µm). Figure 1d shows the TeNT after fs laser irradiation at 25 mW. The yellow circle represents laser-affected areas. It can be noted that there is a critical melting phenomenon on the nanotube. The image shows the thermal effect area is very limited. Melting and solidification only occurred near the end of the nanotube (around 1 µm). Meanwhile, the surface of the non-irradiated area was kept clean and smooth.

During the interaction of an ultrafast pulse with nanomaterials, i.e., Te NWs and silver films, photon absorption and electron stimulation happen within a hundred femtoseconds (fs). This period is shorter than the thermal coupling time between free electrons and lattices, which typically occurs in a couple of picoseconds (ps), depending on the electron-phonon coupling strength of different materials [47]. Therefore, thermal diffusion to the laser-irradiated surrounding area is very limited [48,49]. Our simulation shows that the thermal effect is limited to a few micrometers [50]. The highly localized thermal effect of fs laser irradiation enables the local melting and the accurate joining of Te nanotubes and silver film electrodes and results in a metallic contact. At the same time, the joining process minimizes the effect on the integrity of the nanotubes. A comparison study of continuous wave laser irradiation and fs laser irradiation of a Cu nanowire has been reported in previous reports [50]. Our previous result shows that continuous wave (CW) laser irradiation will cause a significant thermal effect and affect the surface feature of the Cu NW due to the Cu oxidation. However, the effect on the Cu NW by fs laser is very limited [50].

## 3. Results and Discussion

### 3.1. Memristor Behavior Measurement

In this study, quasi-static current-voltage measurements were performed with a Bio-Logic SP-200 electrochemical working station. Figure 2a,d presents the I–V sweeping performances of a TeNT before and after laser irradiation. The pinched loops and symmetrical curves in Figure 2 indicate volatile memristor behavior [51]. When the voltage sweeps from 0 V to 1.5 V, the current response initially remains at a low level. After the voltage reaches a certain threshold value, the current response suddenly increases. This voltage threshold value is referred as the “turn-on” voltage. This behavior shows that the memristor has been switched from HRS to LRS. The device also exhibits a symmetrical behavior for negative sweeps, which indicates that TeNT has a volatile memristor feature at both polarities [52].

It is notable that laser irradiation changes turn-on voltages. Figure 2d,e present enlarged I-V sweep curves of devices before and after laser irradiation. While the main characteristic of the memristor remained similar before and after laser irradiation, the turn-on voltage increased after laser welding. Before laser irradiation, the HRS and the LRS switched at 0.6–0.75 V. After laser irradiation, the switch voltage increased to around 0.85–0.9 V (shown in Figure 2b,d). The turn-on voltage can be shifted by changes of the interface between the TeNTs and the silver electrodes due to either thickening of the diffusion layers or oxidizing of the silver electrodes. A clear relationship between the threshold voltage and the oxidization layer thickness has been reported [53,54,55]. Due to the ion mobility, a thicker oxidization layer provides a higher threshold voltage. Therefore, the turn-on voltage-shift phenomenon probably implies that a thin oxidization layer was introduced by the laser processing.

Figure 2c,f present the current responses of the memristor in the LRS region before and after laser irradiation, respectively. Figure 2c shows that the current responses in the LRS have large variations between cycles. Such large variations could result from a weak contact between the nanotube and electrodes. On the other hand, the stability of the current responses increased after laser irradiation. The increase in stability reflects that a strong mechanical bond and an electrode contact were produced by fs laser irradiation. In the LRS region, the resistances of the non-irradiated sample are around 30–70 kΩ. After laser irradiation, the resistance changed to 40–50 kΩ. Because the resistance of a nanotube is highly dependent on its geometry, it is difficult to provide a critical conclusion as to whether laser irradiation will reduce the conductivity of the nanotube at this point.

Figure 3 shows I-V responses of non-irradiated and laser-irradiated devices under different sweeping frequencies. For the non-irradiated device, the current response shows a significant drop as the frequency increases from 0.001 Hz to 0.1 Hz, indicating that the device needs a long time to fully switch from HRS to LRS. A high-frequency sweep could not fully turn on the device. On the other hand, the current responses for the laser-irradiated device are more stable than the non-irradiated devices as the frequency increases. Figure 3c presents the normalized current response for both types of devices. The figure shows that the current response drops to about 45% of its original value at 0.01 Hz for the non-irradiated sample. For the laser-irradiated sample, the current response stays at 90% of its original value at a frequency of 0.01 Hz. The current response for both samples shows a significant drop at 0.1 Hz. The frequency-dependency of hysteresis is one of the typical memristor characteristics [7]. Overall, the laser-irradiated device shows less frequency-dependency than the non-irradiated device, indicating that the laser-irradiated device has a faster switching speed than the non-irradiated device. There is research showing that laser irradiation can produce heterojunctions and increase mechanical bonding between different types of nanowires [56]. Therefore, we believe that the faster and more stable switching behavior of the laser-irradiated device is due to the better electrical and physical joining bond at the interface between the nanotube and silver electrodes introduced by laser irradiation.

As aforementioned, the device behavior, specifically the resistance, is significantly affected by the nanotube geometry. However, it is very challenging to perform all measurements with identically dimensioned nanotubes, since both the memristor properties and the welding are sensitive to the laser parameters, the nanotube geometry, and the surface roughness of the substrate. In this research, the nanotubes were synthesized by the hydrothermal method, and the geometry control of the nanotubes was very challenging. In order to synthesize uniform nanostructures, the CVD method can offer an alternative approach, but how to transfer and manipulate the nanowire/nanotube remains a big challenge. Furthermore, the surface roughness of printed Ag film is very high, especially at the laser-cut edge (As shown in Figure 1), which leads to a high device-to-device deviation. Therefore, a statistical comparison was performed to compare both types of devices before and after laser irradiation.

Figure 4 presents the calculated resistance of both types of devices at ON status before and after laser irradiation. The resistances are in the range of 100 kΩ to 10,000 kΩ with a large deviation, especially for the non-irradiated samples. Such a significant resistance deviation is mainly due to the unstable contact between the nanotubes and the electrodes. For the non-irradiated device, the nanotube is attached to the electrode by its weight and the surface tension force without any external bonding to enforce the attachment. Additionally, due to the non-uniformity of the electrode surface, the bonding between the nanotube and electrode is very weak. On the other hand, the fluctuation of resistances for the laser-irradiated sample is remarkably improved by comparison with that of the non-irradiated samples. Figure 4b clearly shows that the resistances of the laser-irradiated samples are mainly distributed under 500 KΩ, as the non-irradiated device is under 1500 kΩ.

The theoretical electrical resistivity of Te is between 4.4 mΩ-m and 50 mΩ-m, depending on the crystal structure [57,58]. According to the geometry of TeNT (shown in Appendix A), the estimated theoretical resistances of the TeNT are between 56 kΩ and 600 kΩ. The resistance value of the laser-irradiated devices is in the range of the theoretical resistance value of the TeNTs, which indicates that the effect of contact resistance of laser-irradiated devices is not dominant.

It is noticeable that there is a couple of devices that show a very high resistance value in the non-irradiated devices. These devices are considered as “non-working” devices. Only the devices with resistance values lower than 2000 kΩ, which includes 95% of the data, are used to calculate the average resistance. As a result, the average resistance values of the non-irradiated and laser-irradiated devise are 479.9 kΩ and 278.7 kΩ, respectively. The result shows that laser irradiation reduces the contact resistance between nanotubes and electrodes by enforcing the joining bonding. Moreover, the smaller resistance distribution of laser-irradiated devices indicates that the laser-induced joining increases the device-to-device stability.

To understand the shifting of the turn-on voltage, the corresponding I-V curves were plotted on a semi-logarithmic scale in Figure 5a,b. For the non-irradiated devices, the current response shows a clear voltage dependence, as shown in Figure 5a. The lowest current responses are observed at 0 V and 0.2 V. On the contrary, a significant current drop can be observed for the laser-irradiated device as the voltage increases from 0 V to 1 V (Figure 5b, step 1–step 2). Additionally, the minimum current values are observed to be around ±0.4 V. The current response of the laser-irradiated device shows less voltage dependency than that of the non-irradiated device. It indicates that a weak current remains when sweeping the voltage back to 0 V (point 7), which could be attributed to a capacitive effect introduced by laser irradiation [59]. In general, a capacitive-coupled memristive effect can be observed for metal-oxide memristors [60,61]. The oxidized metal layer performs as an isolation layer and leads to a capacitor-coupled effect of the device. Dash arrows in Figure 5 point out the switching behaviors. As the figure shows, the switching behavior for the non-irradiated device is not obvious. The current response shows a small “jump” around ±0.5 V. For the laser-irradiated device, a critical threshold switching behavior can be observed around ±0.7 V. The threshold switching behavior is most likely to be observed on metal-oxidated material (Table 1). This possibly indicates the formation of an oxidation layer. Based on the Figure 5b, the ON/OFF ratio of a laser-irradiated device is about 10^2^ with 0.5 V reading voltage.

Figure 5c presents a semi-logarithmic scaled current response of another welded Te nanotube with a sweep voltage of ±5 V. The current response shows similar behavior to that seen in Figure 5b. The current response drops in the first stage (step 1) and increases suddenly once the voltage surpasses 0.8 V (step 2). With further increasing voltage, the current response demonstrates a clear voltage dependence and switching of resistive states (steps 2 → 3). The symmetrical phenomenon is observed with backward (from 0 V to 5 V) voltage sweeping (steps 4 → 5 → 6).

Meanwhile, the minimum current response is observed around 1 V and −1.5 V. A similar phenomenon is shown in the −1.5 V to 1.5 V range, as shown in Figure 5b. This phenomenon is attributed to the capacitor-coupled effect between the nanotube and the electrode. At the initial stage of the voltage sweep, the applied voltage charges the coupled capacitor and lowers the current through the nanotube (stage 1). Once the coupled capacitor is fully charged, the current starts to flow along the nanotube, and a clear voltage dependency shows on the current response of the divide (step 2 → 3). When the applied voltage is below 0.8 V, the coupled capacitor starts to discharge and increases the current response (step 3 → 4). As an estimation, based on the equation $i=C\;\left(\frac{dV}{dt}\right),\;$ the coupled capacitance is around 0.56 µF for this sample.

The capacitor-coupled memristor structure is usually applied to a nonlinear circuit design. A passive circuit based on a memristor and coupled capacitors was reported by Ali et al. [62]. The reported circuits can be applied to low-pass, high-pass, and bandpass filters. Semery et al. designed and simulated a fractional-order circuit by placing a 5–10 µF capacitor in series with a memristor [63]. This innovative capacitive-memristive coupled feature may find disruptive application in self-learning, computation, or sensing where the intrinsic capacitance is critical [64,65].

### 3.2. Multistage Characteristic

More importantly, the multilevel current amplification phenomenon has been observed. Figure 6 presents the current response under continuous excitation with forward (0–5 V) voltage sweep cycles. The figure shows that by applying a high forward voltage the current responses were amplified to multiple levels. For the non-irradiated sample, the current response can be amplified for the first four cycles, and the peak current is around 45 µA. However, after the fifth cycle, the current response shows a significant vibration, and the peak current response drops to 20–35 µA. The unstable current response can be caused by unstable contact resistance between the nanotubes and the electrodes. Moreover, an unstable peak current could be due to the high instability of conduction paths under high current flows, since the conductive state depends on the material property and geometry of the device [66]. For the laser-irradiated sample (Figure 6b), the current response can be amplified to four different levels. The amplitude of each amplified level is in a range of 50–70 µA. The current response of the laser-irradiated sample shows higher stability than the non-irradiated sample.

Figure 6c–f demonstrates the multileveled resistance behavior for 0–10 V voltage cycles with a reset voltage of −8 V. As presented in Figure 6b,d, the current response is amplified to different levels depending on different voltage cycle numbers. By comparing before-and-after laser irradiation, it is clear that the current response of the non-irradiated sample is very noisy, and many current spikes could be observed. On the other hand, after laser irradiation, the current response becomes stable, and the noise is reduced significantly. Figure 6c presents that the high current response is reset to the initial level immediately after the bias voltage is applied. This suggests that the resistance state is defined by the previous voltage cycle number, and it appears that the revised bias voltage (−8 V) can erase the previous status and reset a controllable multileveled resistance state. Lin et al. reported a single TiO_2_ nanowire memristor with five resistance levels [39]. The reported device provides a 20–40 nA current response with an 8 V reading voltage. A four-level ZnO nanorod memristor with a 4 V writing voltage and −6 V reset voltage was reported by Russo. The current response of this device is in a range of 2–4 µA, depending on the resistance level [67]. The presented TeNT device in this study works with a 10 V writing voltage and a −8 V reset voltage. By comparing with these metal oxide devices, the TeNT has remarkably high switching currents at the mA level. Table 2 summarizes widely studied single nanowire/nanotube devices. All the devices summarized in the table have a similar structure, where a single nanowire/nanotube bridges two electrodes. By comparison, it is noticeable that TeNT devices have a performance comparable to other materials, which evidence the practicality and effectiveness of using tellurium nanotubes as active materials for memristive devices.

The switching mechanism of a Te-based switching device has been attributed to a potential energy barrier at the Te–electrode interface [68]. A HRS switches to a LRS with a 0.95 eV Schottky barrier. Furthermore, by observing through in-situ TEM, they found that the high ON current was led by voltage pulse–induced crystal-liquid melting transition [69]. The sweeping voltage causes an interface crystal-to-liquid phase change and leads to a volatile ON/OFF switching behavior. At the same time, the resistance of the Te material drops as temperature increases [70]. Te is also known as a good thermal conductor; therefore, the Joule heat created by applying voltage may be involved in the resistance state changes. The multileveled resistance state could be due to oxygen vacancy accumulation, typically observed in metal oxide material [39]. In this study, a thin oxidization layer is possibly formed during laser irradiation. Therefore, the oxygen vacancies from the oxide layer can result in multileveled resistance states. This possible oxide layer can lead to a capacitance behavior of the Te/Ag junction. As reported, the melting point of Te is 449 °C, and that of silver is 961 °C [71]. Depending on the composite ratio, Ag_2_Te or Ag_5_Te_3_ will be generated [71]. Based on the phase diagram provided in Ref. [71], and considering that the melting phenomenon of the silver electrode is very limited (as shown in Figure 1d), the possibility of forming Ag_2_Te is larger than that of Ag_5_Te_3_. However, given the complex nature of laser-induced joining, the detailed mechanism and microstructure characterizations for this system still need further investigation and analysis; such as TEM and EDS can be applied for characterization. The type of junction for Te–Ag and whether a thin oxidation layer formed can be determined by further analyzing the atomic structure and element distribution.

## 4. Conclusions

In this paper, we reported a laser-welded tellurium nanotube memristor on a printed flexible silver film substrate. The devices provide voltage-dependent multifold threshold switching behaviors. A femtosecond laser-induced and localized thermal welding process was developed. Noticeable enhancements of memristor behaviors were observed after laser processing. A capacitor-coupled memristor behavior was discovered and induced by the fs laser joining process, which is possible due to the thin oxidized layer created during the laser irradiation. More importantly, this capacitive-coupling device displayed a multistage resistance state in the HRS. This multileveled resistance state is much stronger than that previously reported for oxide nanowire memristors and can be repeatedly selected by introducing a revised voltage. The voltage performs as a “reset” function. It erases the effect of previous voltage cycles and resets the resistance to the initial status. Although the detailed microstructure characterizations and the underlying physical mechanism need to be elucidated, laser-induced multistage resistance and intrinsically coupled capacitance phenomena provide a very attractive basis for innovative devices in a complex neuromorphic network and smart sensing system.

## Figures and Tables

**Figure 1 nanomaterials-13-00789-f001:**
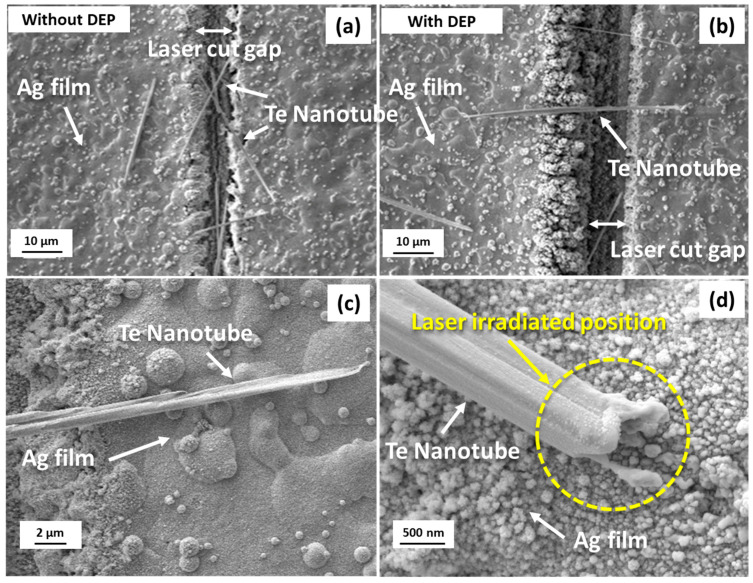
(**a**) SEM images of randomly distributed TeNTs on printed silver film electrodes, (**b**) SEM image of manipulated TeNTs by DEP process, (**c**) SEM images of interface between TeNT and silver film electrodes before fs laser irradiation and (**d**) after fs laser irradiation with 25 mW.

**Figure 2 nanomaterials-13-00789-f002:**
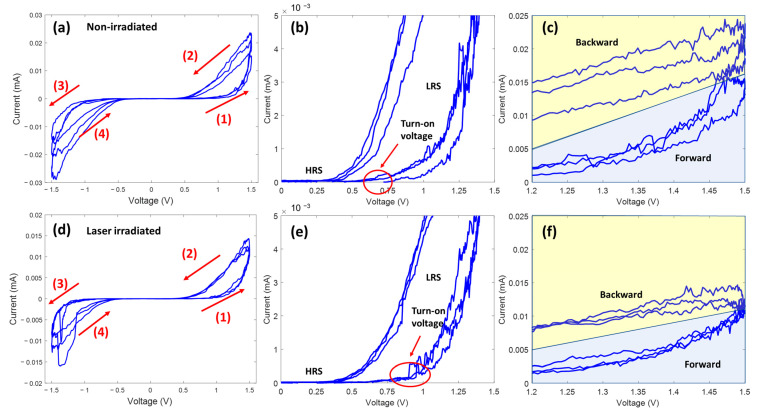
(**a**) I–V curve of TeNT memristors at a sweeping rate of 0.001 Hz, (**b**) enlarged I–V curve with a voltage range of 0–1.5 V, and (**c**) enlarged I–V curve with a voltage range of 1.2–1.5 V. (**d**) I–V curve of laser irradiated TeNT memristors at a sweeping rate of 0.001 Hz, (**e**) enlarged I–V curve with a voltage range of 0–1.5 V, and (**f**) enlarged I–V curve with a voltage range of 1.2–1.5 V.

**Figure 3 nanomaterials-13-00789-f003:**
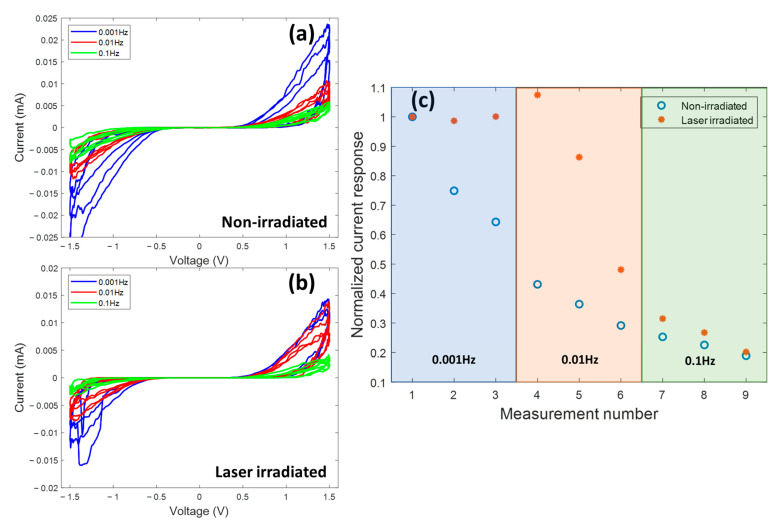
(**a**) I–V curves of a non-irradiated device under different sweeping frequencies, (**b**) I–V curves of a laser-irradiated device under different sweeping frequencies, and (**c**) normalized current responses at the ON stage of non-irradiated and laser-irradiated devices under different frequencies.

**Figure 4 nanomaterials-13-00789-f004:**
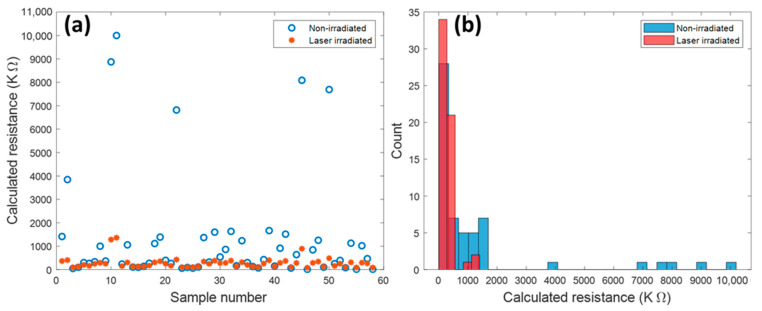
(**a**) Resistance calculated at “ON” status for different samples. (**b**) Statistical result of resistance at “ON” status.

**Figure 5 nanomaterials-13-00789-f005:**
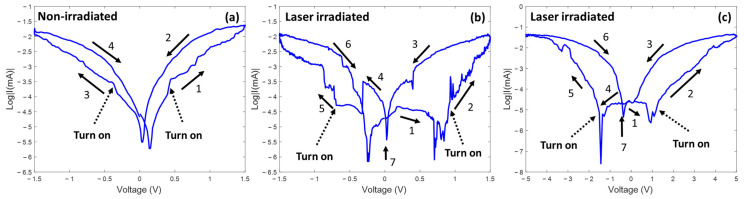
The corresponding I-V curves plotted on a semi-logarithmic scale for (**a**) non-irradiated and (**b**) laser-irradiated samples, and (**c**) semi-logarithmic scaled I-V curves for laser-irradiated sample with a scan range of ±5 V. Dash arrows point out the threshold switching points.

**Figure 6 nanomaterials-13-00789-f006:**
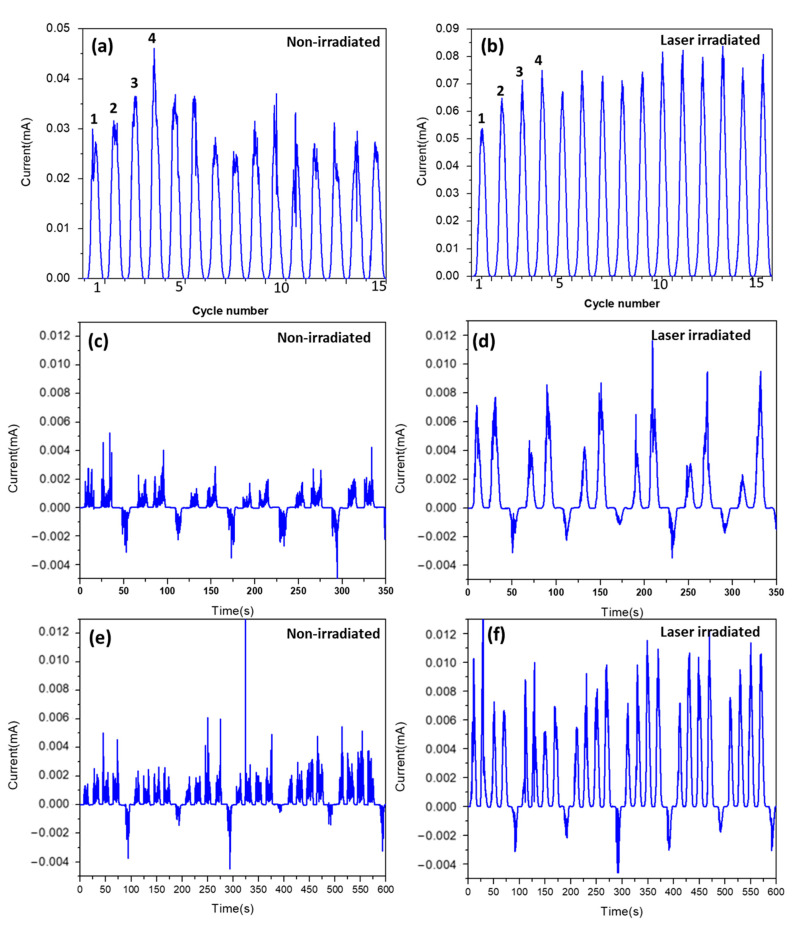
Current response under repeated voltage cycles with sequences of 0 V-5 V-0 V for (**a**) an unirradiated sample and (**b**) a laser-irradiated sample. (**c**–**f**) Current response profiles for the non-irradiated sample and fs laser-irradiated sample. The single forward voltage loop in a sine profile was 0 V-10 V-0 V. The reverse bias used was −8 V.

**Table 1 nanomaterials-13-00789-t001:** A summary of typical volatile memristors.

Devices	Switch Type	Switch Method	ON/OFF Ratio	Ref.
Ag/TiO_2_/Pt	Threshold switching	Filament formation	10^7^	[16]
Cu/SiO_2_/Pt	Threshold switching	Filament formation	10^7^	[17]
Pt/ZnO:Ag/Ti/PET	Threshold switching	Filament formation	10^7^	[18]
Au/MoS_2_/Ag	Threshold switching	Filament formation	10^6^	[19]
Ag/ZrO_2_/Pt	Threshold switching	Filament formation	10^7^	[20]
Pt/CoO/ITO	Threshold switching	Trap-controlled space-charge-limited current	10^2^	[21]
Pt/HfO_2_/TiN	Threshold switching	Electron trapping	10^2^	[22]
W/WO3/PEDOT:PSS/Pt	Analog switching	Migration of protons	10^3^	[23]
ITO/MoO_3_/Ag	Analog switching	Migration of oxygen vacancy	1.28	[24]

**Table 2 nanomaterials-13-00789-t002:** Summary of typical single nanowire/nanotube devices.

Active Material	Electrode Material	NW/NT Diameter	Electrode Spacing	Volatility	|V on|	ON/OFF Ratio	Switching Mechanism	Ref.
NiO	Ti/Au	70 nm	1 µm	N.A ^(a)^	1.2 V	10^3^	Filament formation	[32]
ZnO	Ti/Au	100 nm	500 nm	Nonvolatile	<40 V	10^2^	Migration of oxygen vacancy	[72]
Na-doped ZnO	Ag	300 nm	2.5 µm	Nonvolatile	<40 V	10^2^	Migration of Ag ions	[73]
CuO	Ni	50 nm	2 µm	Nonvolatile	0.87–2.25 V	10^3^	Migration of oxygen vacancy	[74]
HfO_2_/NiO/Ni	Ti/Au	55 +10 + 20 nm ^(b)^	2.8 µm	Nonvolatile	1–3 V	10^7^	Valence changes of Hf_4_^+^	[75]
TiO_2_	Au	200 nm	4 µm	Volatile	5 V	N.A	Migration of oxygen vacancy	[39]
Ag/TiO_2_	Ag/Al	63 +15 nm	2.8 µm	Volatile	0.4 V	10^7^	Migration of Ag ions	[76]
NaxWO_3_	Au	300 nm	4 µm	Volatile	<4 V	N.A	Migration of Na ions	[36]
TeNT	Ag	500 nm/20 nm ^(c)^	5–8 µm	Volatile	0.7–0.9 V	10^2^	N.A	This work

^(a)^ N.A represents that corresponding information has not been provided in the article. ^(b)^ Core-shell structure: nanowire diameter + shell layer thickness. ^(c)^ Nanotube: nanotube diameter/wall thickness.

## Data Availability

Not applicable.

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
