# Peer review of "Femtosecond Laser-Induced Nano-Joining of Volatile Tellurium Nanotube Memristor"

_nanomaterials, 2023, doi:10.3390/nano13050789_

Round 1
Reviewer 1 Report (New Reviewer)
The current manuscript is discussing the Te-nanotube memristor device. Further clarifications are necessary. The comments are as bellow
1. It is not right to claim a memristor. Memristor hysteresis is not the same. Rather this hysteresis is very much like threshold switching.
2. Please provide the device-to-device distribution. The distribution may be very high. How to minimize it.
3. The review of the memristor section is very weak (Electronics 9 (6), 1029, 2020; 2019 J. Phys. D: Appl. Phys. 52 113001; https://www.mdpi.com/2079-9268/12/1/14)
Author Response
Reviewer 1.
The current manuscript is discussing the Te-nanotube memristor device. Further clarifications are necessary. The comments are as bellow
- It is not right to claim a memristor. Memristor hysteresis is not the same. Rather this hysteresis is very much like threshold switching.
Response: Thank you for your comment. As described in the manuscript, the device shows a clear switching behavior and two resistance states: LRS and HRS (Figure 2). We believe that the device shows a typical volatile memristor behavior. The volatile memristors can be divided into two types: digital switching (threshold switching) and analog switching types [1-2]. As our understanding, threshold switching is one of the switching types of the memristor. Also, the relative literature review has been added to the introduction section. (Page 1, lines 54-72)
[1]. Wang R, Yang JQ, Mao JY, Wang ZP, Wu S, Zhou M, Chen T, Zhou Y, Han ST. Recent advances of volatile memristors: Devices, mechanisms, and applications. Advanced Intelligent Systems. 2020 Sep;2(9):2000055.
[2]. Sun, K., Chen, J. and Yan, X., 2021. The future of memristors: Materials engineering and neural networks. Advanced Functional Materials, 31(8), p.2006773.
- Please provide the device-to-device distribution. The distribution may be very high. How to minimize it.
Response: The device-to-device performance distribution highly vibrates for this device, as shown in Figure 4. As a single nanotube-based device, the joining and the memristive performance of the device are very sensitive to the geometry of the nanotube itself. In this research, the nanotubes were synthesized by the hydrothermal method, and controlling the geometry of the nanotubes is very challenging An alternative approach to synthesizing uniform nanostructures would be the CVD method, however, transferring and manipulating individual nanowires or nanotubes remains a significant technical barrier. Furthermore, the surface roughness of printed Ag film is still high, especially at the laser cut edge (Fig. 1). Therefore, the connection between nanotubes and Ag films varies case by case. This leads to a high device-to-device distribution. Improving device-to-device stability will be the focus of future research. The manuscript has been revised based on your suggestions (Page 7, lines 278- 288).
- The review of the memristor section is very weak (Electronics 9 (6), 1029, 2020; 2019 J. Phys. D: Appl. Phys.52113001; https://www.mdpi.com/2079-9268/12/1/14)
Response: Thank you for your suggestion. The literature review part has been modified, and changes are highlighted in the manuscript (Page 1, lines 54-98; Page 3, 111-126).

Reviewer 2 Report (New Reviewer)
See attachment for my feedback

Author Response
Reviewer 2:
Referee report “Femtosecond laser-induced nano joining of tellurium nanotube memristor”; Yongchao Yu, Pooran Joshi, Denzel Bridges, David Fieser and Anming Hun, Nanomaterials-2195800
In the manuscript Yu et al. present their (electrical) characterization results as obtained on non-irradiated and femto-second laser irradiated tellurium nanowire based memristors. The work is certainly of interest and fits with the scope of Nanomaterials, however, some aspects need to be improved. Contents-wise, types/brands/purities/manufacturers of used chemicals and equipment has to be included, in order to be able to reproduce the data (by readers). Moreover, the discussion of Figure 5 needs to be improved since it is unclear in its current state (in fact, it is a bit vague). Language-wise, the authors need to carefully check the (non) use of articles, to improve readability (or consult the English editing service of MDPI). Taking into account these aspects a minor revision is recommended.
Contents
- Section 2 (Materials/Methods): the authors should mention the purities of used chemicals, as well as where it was purchased (and from which company). Moreover, information regarding used equipment, such as type/brand/manufacturer, has to be included (e.g. of the laser, oven, used electronics, etc).. Please comment/clarify.
Response: Thank you for your suggestion. The information on chemicals and equipment has been added to the manuscript (Page 3, section 2.1-2.2).
- Line 110: the authors state that the wall thickness of the nanotubes is about 30-50 nm. How it this range determined? Please add/comment.
Response: The wall thickness was estimated by SEM image and ImageJ software. The relative information was added to the manuscript (Page 4, Line 139).
- Line 124: in which environment (and pressure) was the oven curing done? Please add.
Response: The curing was accomplished in the air at atmospheric pressure. We have modified the manuscript and highlighted the change (Page 4, section 2.2, line 155).
- Lines153/Figure 1c: from the image Fig. 1c cannot be concluded/seen that no physical bonding between the nanotube and silver has occurred: Fig. 1c should show the end of a nanotube, similar to Fig. 1d, as ‘evidence’. Please adapt.
Response: Thank you for your suggestion. We have modified the figure (Page 6, Figure 1).
- Figure 1: the insert of Fig. 1b should be ‘with DEP.’ Please adapt.
Response: Thank you for your correction. We have fixed it (Page 6, Figure 1).
- Line 197: the switch voltage for irradiated samples is certainly not 1V, but 0.85-0.9V. Please adapt.
Response: Thank you for your correction. We have fixed it (Page 6, lines 230).
- Lines 280-317 + Figure 5: the discussion of the shown curves is unclear and a bit vague. Please adapt and it should sound more scientific.
Response: Thank you for your suggestion. We have modified the manuscript to provide a clear description (Page 8, line 278 – Page 9, line 354).
- Line 380: the authors state more ‘microstructure characterizations’ are needed. Which ones, and how should they be applied to the Te-nanotube based memristors? Please specify.
Response: Thank you for your comments. For microstructure characterization, such as TEM and EDS can be applied. The type of junction for Te – Ag and whether a thin oxidation layer formed can be determined by analyzing the atomic structure and element distribution. We have modified the manuscripts according to the suggestion (Page 12, lines 413-420).
- Grammar:
Throughout manuscript: femtosecond should be abbreviated as ‘fs’, not ‘FS’. Please adapt.
ï‚· Throughout manuscript: why are several sentences/sections in blue font? Please adapt.
ï‚· Line 13: “…effect of a tellurium (Te)…”
ï‚· Line 15: “…190 °C. A femtosecond… used to form nanowire…”
ï‚· Line 18: “…between the Te nanotube…”
ï‚· Line 29: “…[1]. Traditional Si-based…”
ï‚· Line 31: “…units. Neuromorphic computation…”
ï‚· Line 41: “Since the HP lab… it has become…”
ï‚· Line 115: “… with a mean diameter …”
ï‚· Line 116: “… and created a dense …”
ï‚· Line 134: “… laser processing system …”
Figure 1: the insert of Fig. 1b should be ‘with DEP’. Please adapt.
ï‚· Line 178: “… electrodes, (b) SEM …”
ï‚· Line 183: “… current-voltage measurements were performed…”
ï‚· Line 202: “… mobility, a thicker…”
ï‚· Line 213: “… it is difficult to…”
ï‚· Line 246: “… with the identically…”
ï‚· Line 271: “As a result…”
ï‚· Line 272: “… of the laser irradiated…”
ï‚· Line 308: “As an estimation, based…”
ï‚· Line 358: “Figure 6. Current response…”
ï‚· Line 365: “… with a 0.95 eV Schottky…”
ï‚· Line 369: “… therefore the Joule heating…”
ï‚· Line 377: Reference [64] is not provided in the list.
Response: Thank you for your very detailed advice. We have modulated the manuscript according to your suggestions. The changes are highlighted in the manuscript.

Round 2
Reviewer 1 Report (New Reviewer)
A comparison table is meaningless without comparing your data. Please update the comparison table with your work. What kind of memory is discussed here? Volatile or non-volatile. The analysis of I-V is insufficient. Please compare the log plot of the I-V and highlight the actual switching. The author must identify the difference between threshold switching and memory. [https://onlinelibrary.wiley.com/doi/10.1002/adfm.201704862; https://onlinelibrary.wiley.com/doi/abs/10.1002/smll.202107575]
Author Response
Reviewer 1:
- A comparison table is meaningless without comparing your data. Please update the comparison table with your work. What kind of memory is discussed here? Volatile or non-volatile.
Response: This Ag/TeNT/Ag device shows a volatile memristor behavior. We have updated our work in table 2, which summiarizes various nanowire/nanotube devices, and relative information has been added to the manuscript. (Lines: 399-405, 433)
- The analysis of I-V is insufficient. Please compare the log plot of the I-V and highlight the actual switching.
Response: Thank you for your suggestion. The actual switching behaviors have been highlighted on the plots and the corresponding description has been added to the manuscript. (Lines 333-339)
- The author must identify the difference between threshold switching and memory.
Response: We do not understanding this question very clearly. Threshold switching refers to the process by which a material changes its electrical conductivity when a certain voltage is applied. As we described in the manuscript, the threshold switching is a type of resistive switch. In 2008, researchers at the HP laboratory reported that a resistive switch could be viewed as a memristive device [1-3]. Therefore, threshold switching is a feature of memristor. As summarized in table 1, a threshold-switching behavior is commonly observed in volatile memristors. Since memory is the ability to store information, the memristors can be classified into two classes based on the memory types, i.e., volatile and non-volatile. A detailed description of volatile and non-volatile memristors has been described in the manuscript (lines 44-53). Besides, in the manuscript, the phrase “multilevel memory” has been changed to “multileveled resistance state” for better clarity, and all the changes have been highlighted.
[1] Pershin, Y.V. and Di Ventra, M., 2011. Memory effects in complex materials and nanoscale systems. Advances in Physics, 60(2), pp.145-227
[2] Lee, J.S., Lee, S. and Noh, T.W., 2015. Resistive switching phenomena: A review of statistical physics approaches. Applied Physics Reviews, 2(3), p.031303.
[3] Strukov, D.B., Snider, G.S., Stewart, D.R. and Williams, R.S., 2008. The missing memristor found. nature, 453(7191), pp.80-83.
- [https://onlinelibrary.wiley.com/doi/10.1002/adfm.201704862; https://onlinelibrary.wiley.com/doi/abs/10.1002/smll.202107575]
Response: Two reference papers have been cited in the manuscript (Ref.14, Ref 31)

Round 3
Reviewer 1 Report (New Reviewer)
No comments
This manuscript is a resubmission of an earlier submission. The following is a list of the peer review reports and author responses from that submission.
Round 1
Reviewer 1 Report
Comments:
The authors described a method of using fs laser nanofabrication in different ways: cutting and joining in single device fabrication, together with electrode printing, to obtain improved performance in resistive switching and multilevel memory application. The methods are novel with solid analysis/comparison. The results and analysis are also with a comprehensive understanding of the mechanism of the laser-matter interaction. The manuscript is suggested to be accepted after revising the following comments:
1. The introduction is suggested to be more focused on the key part of the manuscript. For example, the paragraph on the polymer-based memristors is irrelevant to the manuscript and it is suggested to be deleted. Furthermore, the authors should focus on the nanowire/nanotube-based memristive device and describe why fs laser or laser fabrication is required in these devices by considering the stability and improved interface properties, which could be directly linked to the further discussion in the results section.
2. A few typos existed in the manuscript and should be corrected: Line 149: “NeTN” should be “TeNT”. Line 234: “unusable” should be “unstable”. Line 272: all the lowercase letters “0v”, “8v” and “10v” should be changed to “0V”, “8V” and “10V”.
3. The resulting higher current amplitude in TeNT based devices compared to other metal oxides nanowire-based device should be reconsidered for comparison since for both memory application and neural network applications, lower current amplitude is required to meet the requirement of lower power consumption.
4. In Line 70, it is suggested to delete the wording “storage” as we normally think it should be maintained for sufficient time to be used for storage applications. This is not achieved in the current TeNT device as a volatile performance is demonstrated.
5. For the analysis of the device mechanism, it is unlikely that the electrochemical metallization is involved in the TeNT device as we did not observe a direct abrupt current increase, which could be a sign of filament formation. The authors are suggested to analyze the mechanism in more detail by considering the interface oxidation layer formation and could provide a schematic diagram to describe the potential mechanism.
Author Response
Reviewer 1
The authors described a method of using fs laser nanofabrication in different ways: cutting and joining in single device fabrication, together with electrode printing, to obtain improved performance in resistive switching and multilevel memory application. The methods are novel with solid analysis/comparison. The results and analysis are also with a comprehensive understanding of the mechanism of the laser-matter interaction. The manuscript is suggested to be accepted after revising the following comments:
- The introduction is suggested to be more focused on the key part of the manuscript. For example, the paragraph on the polymer-based memristors is irrelevant to the manuscript and it is suggested to be deleted. Furthermore, the authors should focus on the nanowire/nanotube-based memristive device and describe why fs laser or laser fabrication is required in these devices by considering the stability and improved interface properties, which could be directly linked to the further discussion in the results section.
Response:
Thank you for your comments. The introduction part has been modified regarding the suggestion. We agree with your comments that “polymer-based memristors are irrelevant”. Therefore, this part has been removed from the introduction part. And literature reviews on nanowire/nanotube-based memristors and laser nano-joining have been added to the manuscript on page 2, lines 61 - 79.
- A few typos existed in the manuscript and should be corrected: Line 149: “NeTN” should be “TeNT”. Line 234: “unusable” should be “unstable”. Line 272: all the lowercase letters “0v”, “8v” and “10v” should be changed to “0V”, “8V” and “10V”.
Response:
We have fixed typos.
- The resulting higher current amplitude in TeNT based devices compared to other metal oxides nanowire-based device should be reconsidered for comparison since for both memory application and neural network applications, lower current amplitude is required to meet the requirement of lower power consumption.
Response:
Thank you for your comments. We have modified our manuscript according to your suggestions. Please check highlighted part on page 8, lines 285 -295.
- In Line 70, it is suggested to delete the wording “storage” as we normally think it should be maintained for sufficient time to be used for storage applications. This is not achieved in the current TeNT device as a volatile performance is demonstrated.
Response:
We have revised the description according to your comments. The word “storage” has been deleted, and the sentence has been modified (Page 2, lines: 93-94).
- For the analysis of the device mechanism, it is unlikely that the electrochemical metallization is involved in the TeNT device as we did not observe a direct abrupt current increase, which could be a sign of filament formation. The authors are suggested to analyze the mechanism in more detail by considering the interface oxidation layer formation and could provide a schematic diagram to describe the potential mechanism.
Response:
Thank you for your comments. We have modified our manuscript; The possible switching mechanism of TeNT has been discussed on Page 8, lines 299-314. However, for the TeNT memoriter, to the best of our knowledge, the references are very limited. We added the recent results in Refs. [59-61] and some relevant discussion on page 9. Therefore, at this point, it is challenging to provide a very detailed and confirmed explanation of the switching mechanism. The CV result shows evidence that an oxidation layer may be formed by laser irradiation, but more detailed analysis and observations are required to confirm the conclusion. We would like to study this as our next research topic.

Reviewer 2 Report
The authors reported a joining of tellurium nanotube by femtosecond laser. The nano assembling is interesting where Te nanotube was connected with the silver film under low temperature. Although the final results are somehow good, the figures are tough and some other issues should be addressed.
1. Generally, picosecond laser and nanosecond laser machining work by thermal effect while femtosecond laser is expected to have less heat. From another point of view, how does the femtosecond laser work to connect the nanotube and film? If it is thermal, why “non-thermal” joining was used in the title? If it is not, how does it work?
2. This paper is intended to describe the process of laser-induced joining of nanotube with film. Logically, figures should include the laser-induced process and its device property. However, the SEM images of Te nanotube and its XRD data in Figure 1 damaged the whole organization and pushed the story into chaos. Neither Figure 2 shows the full information. There is no comparison between the unjoined nanotubes and joined nanotubes in properties.
3. The authors claimed with appropriate bias parameters, the nanowires or nanotubes can be transversely aligned to the gap shown in Fig 2(a, b). However, it is unclear how it works and which nanotube they used and connected. More, the scale bar is not unified making the results look odd.
4. Where is the Ag film? The authors should mark it clearly.
5. Why is the phrase red in line 127, page 3?
Author Response
Reviewer 2
The authors reported a joining of tellurium nanotube by femtosecond laser. The nano assembling is interesting where Te nanotube was connected with the silver film under low temperature. Although the final results are somehow good, the figures are tough and some other issues should be addressed.
- Generally, picosecond laser and nanosecond laser machining work by thermal effect while femtosecond laser is expected to have less heat. From another point of view, how does the femtosecond laser work to connect the nanotube and film? If it is thermal, why “non-thermal” joining was used in the title? If it is not, how does it work?
Response:
Thank you for your comments. Indeed, femtosecond laser irradiation induces a very localized thermal effect. To avoid possible misleading, the title has been changed to “Femtosecond Laser-Induced Nano Joining of Tellurium Nano-tube Memristor.”
During the interaction of an ultrafast pulse with nanomaterials, i.e., Te nanowires and silver films, photons' absorption and electrons' stimulation happen within a hundred femtoseconds (fs). This period is shorter than the thermal coupling time between free electrons and lattices, which is typically occurs around a couple of picoseconds (ps), depending on the electron-phonon coupling strength of different materials. Therefore, thermal diffusion to the laser-irradiated surrounding area is very limited. Our simulation shows that such a thermal effect is limited to a couple of micrometers. The highly localized thermal effect of FS laser irradiation enables the local melting and the accurate joining of Te nanotubes and silver film electrodes and results in a metallic contact. At the same time, the joining process minimizes the effect on the integrity of the nanotubes. A comparison study of continuous wave laser irradiation and FS laser irradiation of a Cu nanowire has been reported in previous reports. Our previous results show that CW laser irradiation will cause a significant thermal effect and affect the surface feature of the CuNW due to Cu oxidation. However, the effect on the CuNW by FS laser is very limited. Therefore, FS laser processing is also known as the so called “cold process” or “cold machining.”
Section 2.2 has been modified according to the comments (Page 5, lines 150-165).
- This paper is intended to describe the process of laser-induced joining of nanotube with film. Logically, figures should include the laser-induced process and its device property. However, the SEM images of Te nanotube and its XRD data in Figure 1 damaged the whole organization and pushed the story into chaos. Neither Figure 2 shows the full information. There is no comparison between the unjoined nanotubes and joined nanotubes in properties.
Response:
Thank you for your comments. In this experiment, the Te nanotubes were synthesized by a hydrothermal process in our lab. The detailed synthesis method has been described in Section 2.1, Nanotube synthesis. Therefore, it is necessary to present the geometry of synthesized Te nanotubes and XRD analysis to show that the material of the nanotubes is Te. We appreciate your suggestion, and Figure 1 has been moved to Supporting Material.
In the revised manuscript, Figure 1(c) shows the nanotube without laser irradiation, and Figure 1(d) shows a nanotube after laser irradiation. More specifically, the yellow circle in Figure 1(d) points out the laser-irradiated location. We have modified figures and descriptions in the manuscript on page 4, lines 167-168.
The manuscript is focused comparison of memristor behavior between laser-irradiated and unirradiated devices. The device behavior has been studied and compared in section 3.
- The authors claimed with appropriate bias parameters, the nanowires or nanotubes can be transversely aligned to the gap shown in Fig 2(a, b). However, it is unclear how it works and which nanotube they used and connected. More, the scale bar is not unified making the results look odd.
Response:
The detailed description of the DEP process has been added to the manuscript in section 2.2, page 3, lines 124 - 132.
- Where is the Ag film? The authors should mark it clearly.
Response:
Thank you for your comments. We have modified figures; the Ag film was labeled in Figure 1 (Page 4, line:165).
- Why is the phrase red in line 127, page 3?
Response:
We have fixed the problem. The font color has been changed to black.

Round 2
Reviewer 2 Report
The authors reported a joining of tellurium nanotube with the silver film under femtosecond laser irradiation. Although they have replied one point by point, the paper is written roughly and no significant new theory/mechanism/devices are reported or added after last review. Here below are the details.
Laser-induced Joining of Nanoscale Materials has been reported quite a lot before 2020, including the mechanism of thermal and non-thermal, as well as different materials. From this point, the novelty you claimed is not essential enough for publication in the Nanomaterials journal. (Laser-induced Joining of Nanoscale Materials: Processing, Properties, and Applications, M. Xiao, S. Zheng, D. Shen et al. / Nano Today 35 (2020) 100959; Nanoscale Adv., 2020, 2, 1195–1205)
Suggestions for your publication in the Nanomaterials journal, I think you'd better refer to Nanoscale Adv., 2020, 2, 1195–1205.